# ADAPTIVEDRAG: SEMANTIC-DRIVEN DRAGGING ON DIFFUSION-BASED IMAGE EDITING

## ABSTRACT

Recently, several point-based image editing methods (*e.g.*, DragDiffusion, Free-Drag, DragNoise) have emerged, yielding precise and high-quality results based on user instructions. However, these methods often make insufficient use of semantic information, leading to less desirable results. In this paper, we proposed a novel mask-free point-based image editing method, **AdaptiveDrag**, which provides a more flexible editing approach and generates images that better align with user intent. Specifically, we design an auto mask generation module using super-pixel division for user-friendliness. Next, we leverage a pre-trained diffusion model to optimize the latent, enabling the dragging of features from handle points to target points. To ensure a comprehensive connection between the input image and the drag process, we have developed a semantic-driven optimization. We design adaptive steps that are supervised by the positions of the points and the semantic regions derived from super-pixel segmentation. This refined optimization process also leads to more realistic and accurate drag results. Furthermore, to address the limitations in the generative consistency of the diffusion model, we introduce an innovative corresponding loss during the sampling process. Building on these effective designs, our method delivers superior generation results using only the single input image and the handle-target point pairs. Extensive experiments have been conducted and demonstrate that the proposed method outperforms others in handling various drag instructions (*e.g.*, resize, movement, extension) across different domains (*e.g.*, animals, human face, land space, clothing). **The code is provided in the supplementary materials**.

## 1 INTRODUCTION

Benefiting from the huge amount of training data and the computation resource, diffusion models developed extremely fast and derived plenty of applications. For example, the text-to-image(T2I) diffusion model Saharia et al. (2022) attempts to generate images with the input text prompt condition. However, constraining the generation process in this way is often unstable, and the text embedding may not fully capture the user's intent for image editing.

In order to realize fine-grained image editing, previous works are usually based on GANs methods Abdal et al. (2019) with latent space, such as the StyleGAN utilizes the editable $\mathcal{W}$ space.

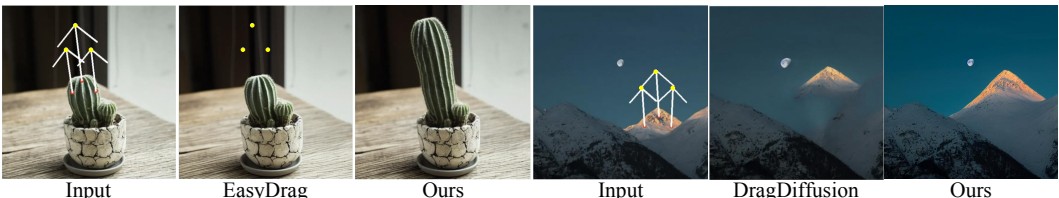

| Input | EasyDrag | Ours | Input | DragDiffusion | Ours |

Figure 1: Existing methods face two main issues: (a) 'Drag missing' (left): EasyDrag fails to guide the succulent to the target points because the point search is ineffective during long-scale drag instructions. (b) 'Feature maintenance failure' (right): DragDiffusion fails to maintain the feature in the middle part of the mountain when the peak is dragged to a higher position.

Recently, DragGAN Pan et al. (2023) introduced a point-to-point dragging scheme to edit images, providing a way to achieve fine-grained content change.

Due to the shortcomings of GAN methods Abdal et al. (2019) in terms of generalization and image quality, the diffusion model Ho et al. (2020) was proposed, offering improved stability and higher-quality image generation. DragDiffusion Shi et al. (2024b) first adopts the point-to-point drag scheme from DragGAN Pan et al. (2023) on the diffusion model. It employs LoRA Hu et al. (2021) to maintain the consistency between the original image and results, then optimizes the latent via motion supervision and point tracking steps. However, the update strategy for point-based drags in DragDiffusion has several limitations, making it challenging to achieve satisfactory editing results. Firstly, users must use a brush to draw a mask, defining the area they wish to adjust. This not only increases the operational complexity of image editing but also makes the results sensitive to the mask region. EasyDrag Hou et al. (2024) simplifies the operation by generating the mask area via the normalized gradients over the threshold $g$. However, the gradients of the entire image are not directly related to the user's edit points, and more critically, the disappearance or cumulative error of these gradients might often result in significant distortions. Another weakness is the fixed step and feature updating region strategy in the latent optimization. For varying dragging distances, the fixed number of iterations cannot effectively optimize the latent representation to reach the target points, leading to the issue of **'Drag Missing'** (left side of Fig. 1). As mentioned in Liu et al. (2024), when the feature differences in the neighboring areas are minimal, the **'Feature maintenance failure'** occurs. However, fixed feature updating regions inevitably blend with surrounding features together, leading to increased similarity with adjacent areas. As a result, in the right part of Fig. 1, existing methods fail to preserve the features at the center of the mountains during long-scale editing.

In this paper, we introduce a novel point-based image editing approach called **AdaptiveDrag** to address the aforementioned issues. *(1) Auto Mask Generation.* We propose an auto-mask generation scheme that integrates both image content and drag point positions. To better align the image content with the mask, inspired by Mu et al. (2024), we get the image elements by the Simple Linear Iterative Clustering (SLIC) Achanta et al. (2012). It segments the image into patches on the feature space of the Segmentation Anything Model 2 (SAM 2) Ravi et al. (2024). Next, We propose a line-searching strategy to generate the final mask, informed by the positions of the handle points and target points. Ultimately, this process automates the generation of a mask that precisely covers the area to be edited and aligns with the user's intent. *(2) Semantic-Driven Optimization.* We incorporate semantic relative information into our latent optimization. Specifically, we designed a position-supervised backtracking strategy to enable adaptive step iteration, effectively handling different drag lengths. For feature region selection, we use segmentation patches from the SLIC results, providing a more precise area for motion supervision and point tracking steps. *(3) Correspondence Sample.* To address the instability of the sampling process, our method incorporates a corresponding loss function between the regions of handle points and target points. Finally, our proposed method can effectively generate high-quality images based on a variety of user drag instructions.

In summary, our contributions to this paper are as follows:

- We propose a mask-free drag method, called **Auto Mask Generation**, via semantic-driven segmentation to automatically generate a precise mask area. It offers users an easy-to-operate but accurate approach to image editing without explicitly drawing the user mask.

- We design an adaptive strategy for the latent optimization process, called **Semantic-Driven Optimization**. It employs a semantics-driven automated process for managing drag steps, update regions, and update radius. Coupled with the adaptive strategy, this approach yields drag results that are more aligned with the semantic features of the input image and compatible with the target points.

- We propose **Correspondence Sample** to improve the generation stability of the diffusion process, encouraging the semantic consistency between regions of handle and target points.

Extensive experiments have been conducted, demonstrating that our AdaptiveDrag outperforms existing approaches in handling a variety of drag instructions (*e.g.*, resize, movement, extension) and across different domains (*e.g.*, animals, human face, land space, clothing).

## 2 RELATED WORK

### 2.1 GAN-BASED IMAGE EDITING

Interactive image editing involves modifying an input image based on specific user instructions. Existing control methods, which rely on text instructions Brooks et al. (2023); Lyu et al. (2023); Meng et al. (2021) and region masks Lugmayr et al. (2022), suffer from precision issues, while image-based referencing methods Chen et al. (2024b); Yang et al. (2023) fall short in terms of control flexibility. Point-based image editing employs a series of user-specified handle-target point pairs to adjust generative image content, aligning with target point positions. For instance, Endo Endo (2022) introduces a latent transformer to learn the connection between two latent codes using Style-GAN Mokady et al. (2022). DragGAN Pan et al. (2023) proposes an updating scheme involving "point tracking" and "motion supervision" within the feature map to align handle points with their corresponding target points. However, GAN-based methods often struggle with complex instructions and yield unsatisfactory results due to their limited model capacity.

### 2.2 DIFFUSION-BASED IMAGE EDITING

Recently, the impressive generative capabilities of large-scale text-to-image diffusion models have led to the development of numerous methods based on these models Rombach et al. (2022); Saharia et al. (2022). For interactive image editing, DragDiffusion Shi et al. (2024b) employs a point-based image editing scheme based on the diffusion model, similar to DragGAN. This method utilizes LoRA for identity-preserving fine-tuning and optimizes the latent space using the loss function of motion supervision and point tracking. However, as shown in Fig. 1, previous methods (*e.g.*, DragDiffusion, EasyDrag) face two main issues: 'drag missing' and 'feature maintenance failure' which result in the latent being incorrectly positioned in certain regions. FreeDrag Ling et al. (2024) introduces a template feature through adaptive updating and line search with backtracking strategies, resulting in more stable dragging. DragNoise Liu et al. (2024) presents a semantic editor that modifies the diffusion latent in a single denoising step, leveraging the inherent bottleneck features of U-Net. Nevertheless, these methods still have challenges when dragging over long distances or across complex textures. To design a user-friendly point-based image editing method, Easy-Drag Hou et al. (2024) leverages gradients in the motion supervision process that remain unchanged in areas with small gradients, and it automatically generates the mask $\mathbf{M}$. Moreover, some methods Shi et al. (2024a); Shin et al. (2024); Lu et al. (2024); Cui et al. (2024) attempt to improve the quality of results in various ways (*e.g.*, drag by regions Lu et al. (2024), flow-based drag Shin et al. (2024), and the fast editing method Shi et al. (2024a)). However, these previous methods provided the mask is not always directly related to the image content, which can result in inaccurate mask generation and unsatisfactory image outcomes. In contrast to previous work, we propose a novel semantic-driven point-based image editing framework that achieves precise results across different drag ranges without the need for a mask.

## 3 METHOD

### 3.1 PRELIMINARY ON DIFFUSION MODELS

Denoising diffusion probabilistic models (DDPM) Ho et al. (2020); Sohl-Dickstein et al. (2015) are generative models that map pure noise $\mathbf{z}_T$ to an output image $\mathbf{z}_0$, using a conditioning prompt to guide the noise prediction process. During the training process, the diffusion model updates the network $\epsilon_\theta$ to predict the noise $\epsilon$ from the latent $\mathbf{z}_t$:

$$\mathcal{L}_\theta = \mathbb{E}_{z_0, \epsilon \sim N(0,I), t \sim U(1,T)} \|\epsilon - \epsilon_\theta(z_t, t, \mathcal{C})\|_2^2, \tag{1}$$

where the sample $\mathbf{z}_t$ is from $\mathbf{z}_0$ with adding noise $\epsilon$. Moreover, the $\epsilon$ is according to the diffusion step $t$ and the condition of $\epsilon_\theta$ $\mathcal{C}$. In the inference process, we employ DDIM Song et al. (2020) for sampling, which reconstructs the target images:

$$z_{t-1} = \sqrt{\frac{\alpha_{t-1}}{\alpha_t}} z_t + \sqrt{\alpha_{t-1}} \left( \sqrt{\frac{1}{\alpha_{t-1}} - 1} - \sqrt{\frac{1}{\alpha_t} - 1} \right) \epsilon_\theta(z_t), \tag{2}$$

where the $\alpha_t$ $(t = 0, 1, ..., T)$ represents the noise scale in each step.

**DDIM inversion** The ODE process can be inverted within a limited number of steps, mapping the given image to the corresponding noise latent:

$$z_{t+1} = \sqrt{\frac{\alpha_{t+1}}{\alpha_t}} z_t + \sqrt{\alpha_{t+1}} (\sqrt{\frac{1}{\alpha_{t+1}} - 1} - \sqrt{\frac{1}{\alpha_t} - 1}) \epsilon_\theta(z_t), \tag{3}$$

**Stable Diffusion** Stable Diffusion (SD) Rombach et al. (2022) is a large-scale text-image generation model that compresses the input image into a lower-dimension latent space using Variational Auto-Encoder (VAE) Kingma (2013). In this study, we base our model on the Stable-Diffusion-V1.5 framework. By extending the DragDiffusion approach, we fine-tune the diffusion model using LoRA Hu et al. (2021), which significantly enhances the diffusion U-Net's capability to more accurately preserve the features of the input image.

## 3.2 OVERVIEW

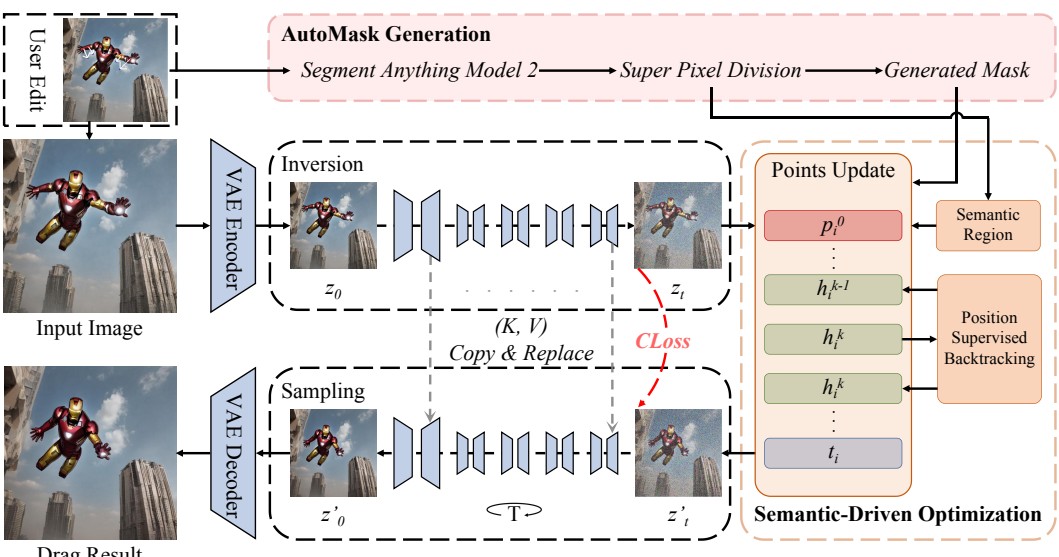

Figure 2: The overall framework of **AdaptiveDrag** comprises four key steps: diffusion model inversion, auto mask generation, semantic-driven optimization, and correspondence sample. Firstly, the model obtains the noised feature $z_t$ through inversion and generates the mask using the auto mask generation module. Secondly, the semantic-driven optimization updates $z_t$ based on the handle point $p_i^0$ and the target point $t_i$ specified in the user's instructions. Thirdly, we perform the sampling operation to denoise $z_t'$ using reference-latent-control $(K, V)$ and the corresponding feature alignment loss $(CLoss)$ on $z_t'$. Finally, we obtain the drag result from the $z_0'$, as predicted by DDIM sampling.

Our AdaptiveDrag aims to achieve two objectives: to flexibly modify the image and to generate accurate and feature-preserving results. The overall framework of our method, illustrated in Fig. 2, is built upon a pre-trained Stable-Diffusion-V1.5 model. The improved modules we propose are color-coded in the figure for clarity. We give detailed descriptions of our method as follows: (1) We introduce the **Auto Mask Generation** module in Sec. 3.3, designed to facilitate more flexible editing. (2) In Sec. 3.4, we describe the **Semantic-Driven Optimization**, which includes the adaptive drag step and the semantic drag region to better explore the context features. (3) Finally, the **Correspondence Sample** is introduced in Sec. 3.5 to mitigate the instability of the sampling process in diffusion and to maintain consistency in the handled regions between input and output images.

## 3.3 AUTO MASK GENERATION

For a user-friendly point-based image editing method, users should focus solely on which image they are editing and the position they wish to modify. Previous methods Shi et al. (2024b); Mou et al. (2023) require a user-input mask to define the regions for content changes, which can be

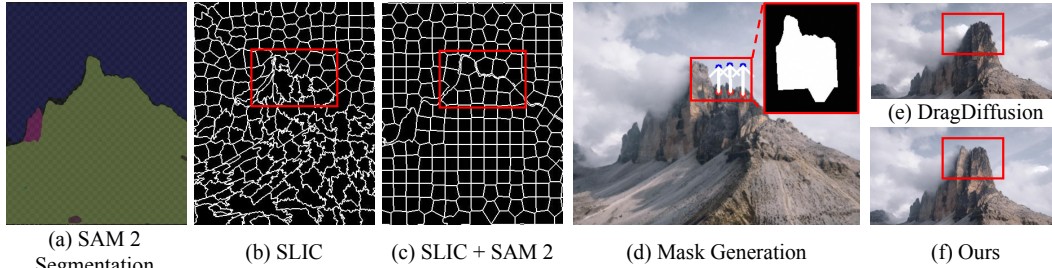

(a) SAM 2 Segmentation    (b) SLIC    (c) SLIC + SAM 2    (d) Mask Generation    (e) DragDiffusion    (f) Ours

Figure 3: Results of different segmentation schemes. (a) The SAM 2 Ravi et al. (2024) segmentation result for the landscape view, effectively separating the overall mountain from its surroundings. (b) The super-pixel patches generated by the SLIC algorithm in the RGB space of the input image, appear chaotic. (c) The result of applying SLIC in the feature space of SAM 2, reveals a clearer and more finely divided representation of the mountainous region. (d) The auto mask generated when the user drags upward from the peak area. (e) / (f) The drag results of DragDiffusion and ours show that the proposed approach achieves a more precise positioning while preserving the original features of the mountain, effectively avoiding the mixing of the two peaks.

cumbersome to operate and may mislead the latent optimization. EasyDrag employs a gradient-based mask generation network. However, it still faces challenges with "drag missing" during long-range drags due to gradient vanishing. To create an auto mask generation module that aligns more effectively with image content, we design a super-pixel mask generation scheme.

As shown in Fig.3 (a), we first use the Segment Anything Model 2 (SAM 2) Ravi et al. (2024) to obtain the segmentation result of the input image. However, we found that SAM 2 primarily focuses on the overall object (*e.g.*, the mountain), often segmenting it into a single patch. This limitation makes it challenging to drag only specific parts of the mountain, such as sections of the peak while preserving the rest. Next, we introduce Simple Linear Iterative Clustering (SLIC) Achanta et al. (2012) to achieve more fine-grained segmentation. However, directly employing SLIC on the RGB space of the image will produce irregular and chaotic results (Fig.3 (b)). **To this end, to get segmentation regions that are semantically consistent within itself while also having fine-grained differences from adjacent areas, we instead employ the SLIC method on the output feature space of SAM2 to achieve a more accurate division of semantic super-pixel patches.** Based on the super-pixel patch division from SLIC, we first select the relevant patches associated with the handle points to form an initial area. Then, we extend the area along the line connecting each handle and target point, and finally, we generate the full mask region for the drag operation. For example, we present the mask result of the peaks with an upward drag operation in Fig. 3 (d) which retains the same edges as the mountains. Since the more precise mask guidance is provided, our method allows for accurate dragging without conflating multiple peaks, as illustrated in Fig. 3 (e) / (f).

## 3.4 SEMANTIC-DRIVEN OPTIMIZATION

Building on the inversion stage and the automatic mask generation module, we propose a novel semantic-driven optimization that enhances the precision of image editing by improving the correlation between the input images and instructions, ensuring the edits more accurately align with the image's context. Following a similar design to DragGAN and DragDiffusion, the main latent optimization process in our proposed method also consists of two key steps: motion supervision and point tracking, which are implemented consecutively. Next, the two steps are then repeated iteratively until all handle points reach their respective targets. As illustrated in the orange box of Fig 2, the design optimization module consists of two parts. First, for the repeated steps, we propose position-supervised backtracking, as detailed in Sec. 3.4.1, to adaptively adjust the number of steps based on the positions of input drag points and predicted points in each point tracking step. The other component is the semantic region, described in Sec. 3.4.2, which is used to constrain the feature area during the motion supervision and point tracking steps. It leverages super-pixel patches from the auto mask generation to form regions that more accurately align with the image content.

### 3.4.1 POSITION SUPERVISED BACKTRACKING

Assuming we are performing the $k$-th iteration to edit the input image, it is crucial to ensure that each step moves toward the appropriate position in the optimization process, effectively guiding it to reach the corresponding target point $t_i$. We focus on two optimization aspects: 1) The direction toward the target points, and 2) The

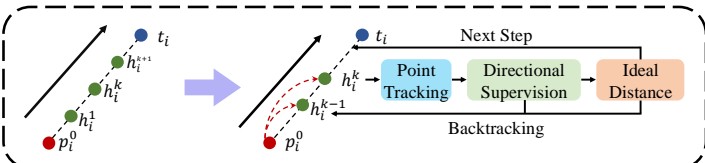

Figure 4: Illustration of our position supervised backtracking pipeline. $p_i^0$, $h_i^k$, $t_i$ denote the handle point, the current searching point in $k$-th updating, and the target point, respectively. The left side illustrates the standard optimization process, while the right side presents our backtracking design, which incorporates both the moving direction and moving distance into the constraints of point optimization.

appropriate number of steps based on varying dragging distances. Specifically, moving in the wrong direction can result in repetitive and ineffective drag updates between the handle point and corresponding targets, preventing the updated point $h_i^k$ from reaching the desired position. Moreover, using a fixed step count for updates as a hyperparameter (*e.g.*, DragDiffusion employs 80 steps) may not be optimal for either small- or large-scale editing. We propose a position supervision backtracking scheme to address the aforementioned issues, as illustrated in Fig. 4. First, we detect the angular relationship between the update point $h_i^k$ and the previous one $h_i^{k-1}$, employing the cosine angle formula to compute the angle between the line connecting $p_i^0$ and $t_i$. We retain the update step only if it has a positive value, indicating movement toward $t_i$. Furthermore, to address the issue of a fixed step number, we introduce a backtracking mechanism. Concretely, we evaluate the moving distance in each step. We define the ideal distance $d = l/n$, where $l$ represents the length from $p_i^0$ to $t_i$ and n denotes the user-defined number of steps. Then, we consider two cases: In the first case, if a suitable optimization occurs where $h_i^k$ reaches the distance $d$, we retain this step. In the second case, if the feature dragging within a step is insufficient, we continue the optimization at the current point by reusing $h_i^k$ as $h_i^{k-1}$ and incrementing the step count. To prevent the optimization from getting stuck in a loop, we introduce a maximum number of updates, denoted as $n_{max}$. By combining the two designs described above, we achieve position supervised backtracking, which ensures that the update process adapts to varying directional and distance instructions.

### 3.4.2 SEMANTIC REGION

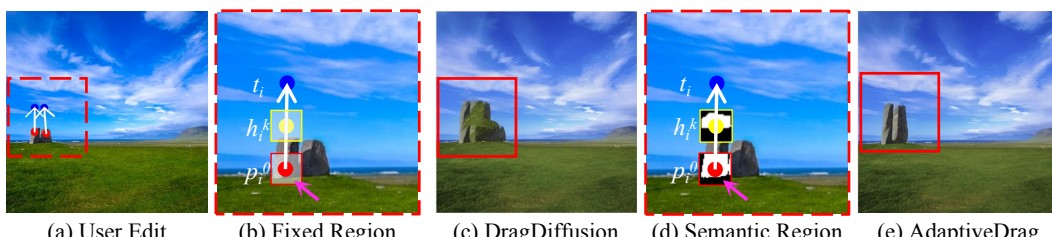

| (a) User Edit | (b) Fixed Region | (c) DragDiffusion | (d) Semantic Region | (e) AdaptiveDrag |

Figure 5: Illustration of the semantic-driven feature optimization where the red, yellow, and blue points represent the handle, predict, and target points, separately. (a) The input image with user instructions. (b) The point tracking process utilizes a fixed square patch (red box) that includes additional grass features (indicated by the pink arrow). (d) The semantic region design provides a more precise mask for the patch, as illustrated in the red and yellow boxes. (c) / (e) Visual comparison: DragDiffusion employs a fixed square region with length $r$, where the grass features are mixed with the stone. In contrast, our approach produces a clearer dragging result based on the semantic region.

As shown in Fig. 5 (a), our goal is to make the giant stones taller. In the point updating process of DragDiffusion (Fig. 5 (b)), $p_0$ serves as the handle point, and the next point $h_i^k$ is predicted through motion supervision and point tracking steps. However, it performs these two steps within a square area (red and yellow boxes) with a fixed side length $r$. This can easily result in the predicted points not being consistently tracked in alignment with the direction of the target points. Once the tracked point is not guaranteed, it can destabilize the update process and ultimately lead to the failure of the drag instruction. For example, in Fig. 5 (c), although the rock became taller, numerous green mounds of grass appeared on it. This occurs because the fixed square update region cannot distinguish between the features of the dragged object and those of other elements, resulting in a mix

of grass and stone features and producing outcomes that do not align with the user's expectations. To address this issue, we propose a semantic region to achieve a cleaner updating area. Specifically, we use the patch region divided by the super-pixel division (as described in Sec. 3.3) for the two updating steps. As shown in Fig. 5 (d), we replace the red and yellow square patches with two semantic super-pixel masks in our semantic-driven optimization. These semantic-driven regions provide adaptive areas that allow our update process to achieve precise and desirable results without being influenced by surrounding elements. Finally, as illustrated in Fig.5 (e), our method using the semantic region achieves a higher quality result that aligns with the user's instructions.

## 3.5 CORRESPONDENCE SAMPLE

Due to the aforementioned designs focusing on optimizing the initial latent, the sampling process in diffusion still lacks adequate control during noise prediction. We observe that when editing an object from red point A to blue point B, the optimal result is achieved when the region around point B in the output image closely resembles the area surrounding point A. As illustrated in Fig. 6, we introduce the Corresponding Loss (CLoss) during the sampling of our point-based image editing framework.

Specifically, CLoss computes the patch $p_A$ around the handle point from $z_0$ (red box) and the target area $p_B$, extracted from $z_0'$, where $z_0$ is the initial noised latent and $z_0'$ is the predicted latent output from the U-Net. In detail, CLoss is a contrastive loss based on symmetric cross entropy Radford et al. (2021), designed to maximize the cosine similarity between $p_A$ and $p_B$:

$$CLoss = \sum_i \mathbb{E}(p_{iA}, p_{iB}), \qquad (4)$$

where $p_{iA}$ and $p_{iB}$ represent the patches from the $i$-th handle and target point, respectively. $\mathbb{E}$ denotes the symmetric cross-entropy loss.

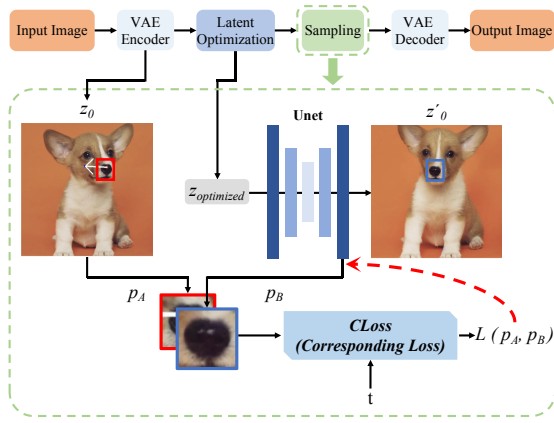

Figure 6: The scheme of correspondence sample.

## 4 EXPERIMENTS

### 4.1 IMPLEMENTATION DETAILS

In experiments, we implement our methods using the Stable-Diffusion-V1.5. Following the DragDiffusion, our method employs LoRA in the attention module for identity-preserving fine-tuning, with the rank as 16. We use the AdamW optimizer Kingma & Ba (2015) for LoRA fine-tuning with a learning rate of $5 \times 10^{-4}$ and a batch size of 4, over 80 steps. During the inference process, we use DDIM sampling with 50 steps, optimizing the latent at the 35th step. We also do not use the classifier-free guidance (CFG) Ho & Salimans (2022) in the DDIM sampling and inversion process. The maximum initial optimization step is 300.

### 4.2 QUALITATIVE EVALUATION.

We perform visual comparisons using the DragBench dataset Shi et al. (2024b), which includes 211 diverse types of input images, corresponding mask images, and 394 pairs of dragging points. Comparing the proposed AdaptiveDrag with other three state-of-art methods: DragDiffusion Shi et al. (2024b), DragNoise Liu et al. (2024) and EasyDrag Hou et al. (2024), we present the visual results shown in Fig. 7. In particular, our method achieves a superior performance of dragging precision and feature maintenance even with small- or large-scale manipulations, where ordinary methods typically falter. For example, the first row in Fig. 7 demonstrates that AdaptiveDrag successfully rotates the large vehicle while preserving the car's basic shape, structure, and position relative to the surrounding scenery. However, DragDiffusion and DragNoise incorrectly position the wheels, while EasyDrag fails to preserve the car's basic structure.

As shown in the second row of Fig. 7, the proposed method demonstrates superior quality compared to the others when modifying different parts of the image through multi-point dragging. The user

| User Edit | DragDiffusion | DragNoise | EasyDrag | AdaptiveDrag (Ours) |
|---|---|---|---|---|

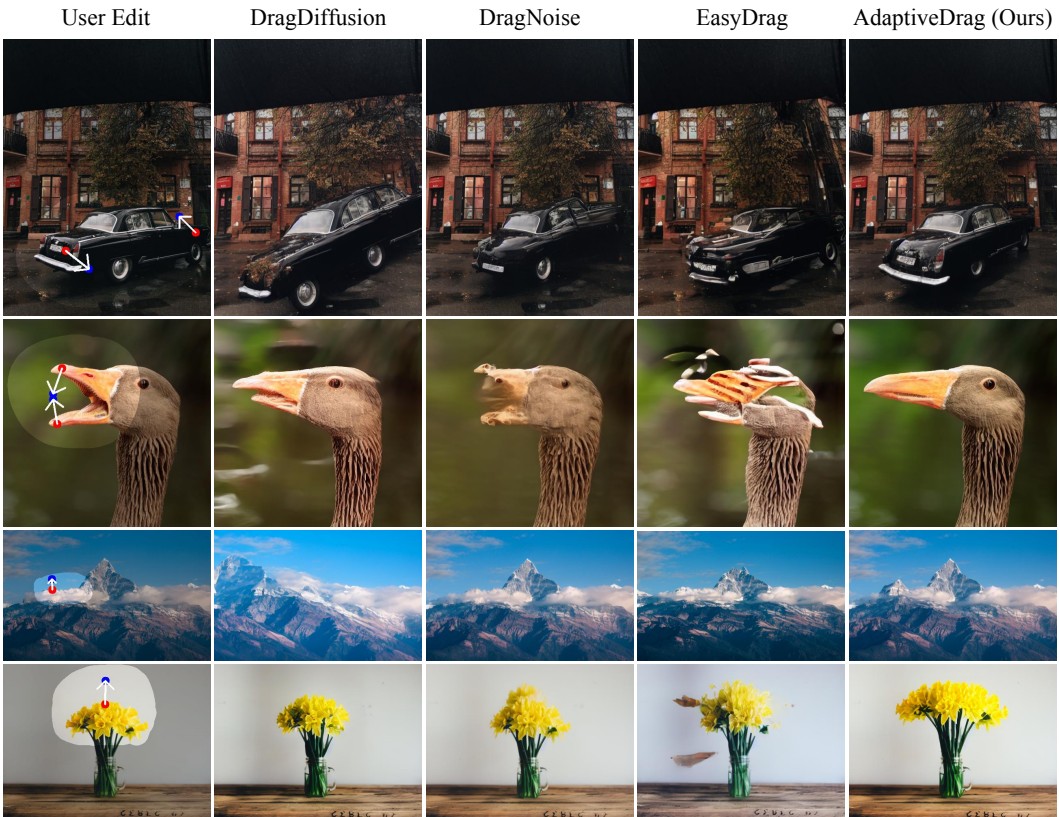

Figure 7: Visual Comparison with other state-of-art methods based on the DragBench dataset. Our method delivers more precision and high-quality results. Notably, the masks shown in the left column are only utilized by the comparison methods.

instruction aims to close the duck's mouth, but DragDiffusion leaves a small gap between the beaks, while the other two methods fail to preserve the basic features of the duck's head. In contrast, our method successfully generates a closed mouth, accurately moving the beaks to the desired position.

Additionally, we apply our method to tiny scale editing, as shown in the last two rows of Fig. 7. In the third row, the objective is to drag a small peak, hidden in the clouds, to a higher position. However, all three compared methods fail to move the corresponding peak from the handle point while AdaptiveDrag accurately identifies the correct region and generates a peak that precisely aligns with the target point's location. The last row illustrates the results of editing a flower which has a complex texture structure. Although DragNoise and EasyDrag move the top of flowers to a higher position, they still fail to maintain a natural growth pattern, altering only the area around the handle points. Compared to the other methods, our result is more consistent with real-world semantic information and aligns more accurately with the user's intent. Additional visual comparisons can be found in Appendix A.1 and more results are illustrated in Appendix A.2, where we conduct further experiments on **rotation, movement, multi-point adjustments, and long-scale editing operations**, separately.

### 4.3 QUANTITATIVE EVALUATION.

To better demonstrate the superiority of our proposed method, we conduct a quantitative comparison using the DragBench dataset Shi et al. (2024b) to illustrate the effectiveness of our approach. For the comparison metrics, we adopt the mean distance (MD) Pan et al. (2023) and image fidelity (IF) Kawar et al. (2023). Especially, the MD calculates the distance between the dragged image and target points to assess the precision of the editing and the IF represents the similarity between the user input image and the results using the learned perceptual image patch similarity (LPIPS) Zhang et al. (2018). In our comparison, the values of IF are calculated as 1-LPIPS.

Table 1: Quantitative evaluation with state-of-art methods on the DragBench Shi et al. (2024b) dataset. Lower MD metrics indicate more precise drag results, and higher IF (1-LPIPS) signifies better similarity between the generated results and the user-edited images. All experiments are conducted on a single Nvidia V100 GPU.

|  | DragDiffusion Shi et al. (2024b) | DragNoise Liu et al. (2024) | EasyDrag Hou et al. (2024) | AdaptiveDrag (Ours) |
|---|---|---|---|---|
| Conference | CVPR 2024 | CVPR 2024 | CVPR 2024 | - |
| MD ↓ | 34.29 | 40.89 | 34.44 | **30.69** |
| IF (1-LPIPS) ↑ | 0.789 | 0.861 | **0.882** | 0.873 |

As shown in Tab. 1, we present the quantitative result of AdaptiveDarg using the two aforementioned metrics. Compared with three state-of-art methods, *i.e.*, DragDiffusion Shi et al. (2024b), DragNoise Liu et al. (2024) and EasyDrag Hou et al. (2024), where the DragDiffusion serves as the baseline for our method, the EasyDrag is the first mask-free point-based image editing framework. AdaptiveDrag achieves the best score in the MD metric when compared to other state-of-the-art methods. It significantly outperforms the previous leading method, DragDiffusion Shi et al. (2024b), with a notable improvement of 3.60, which corresponds to a 10.5% enhancement.

In terms of the IF metric, our method achieves the second-best score, surpassing the baseline DragDiffusion by 0.084, which represents an 8.4% improvement. Although EasyDrag achieved the best IF score, this may be attributed to the occurrence of 'drag missing'. In the visual comparison shown in Fig. 8, we present the input image and results. However, the generated image from EasyDrag has a lower IF score, yet the position of the bow remains unchanged (refer to the red line for comparison). Our method demonstrates the improved editing of the ship.

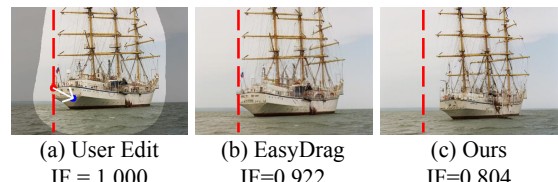

| (a) User Edit | (b) EasyDrag | (c) Ours |
|---|---|---|
| IF = 1.000 | IF=0.922 | IF=0.804 |

Figure 8: An extra explanation of the IF metrics, highlighting the comparison between EasyDrag and our method. (a) The user inputs the image and editing instruction, achieving the highest IF score of 1.0. (b) The generated result of EasyDrag which achieves a higher IF score but fails to move the bow to the target position. (c) Our method successfully drags the bow away from its original position (indicated by the red line).

### 4.4 GENERALIZATION

In addition to the experiments conducted on the standard DragBench benchmark, we performed more dragging experiments using images from various other scenarios to demonstrate the generalizability of our approach. Inspire from the fashion design Baldrati et al. (2023); Kong et al. (2023); Xie et al. (2024) task and try-on Chen et al. (2024a); Zhu et al. (2023); Kim et al. (2024) task, We applied our method to fashion clothing images from the VITON-HD dataset Choi et al. (2021). It contains 13,679 high-resolution virtual try-on images, featuring upper garments, lower garments, and dresses. As shown in Fig. 9, we present the results of point-based image editing applied to clothing. In particular, we generated the editing results with **mask-free** operation, relying only on the input of handle-target point pairs.

For instance, in the first row of Fig. 9, our method enables directional adjustments to clothing, such as elongating sleeves, increasing the coverage area of upper garments, and lowering the height of pants. The proposed AdaptiveDrag modifies the clothing on models while maintaining the body posture (*e.g.*, arm length, shoulder position) and preserving the basic features of the clothing (*e.g.*, sleeve shape). Moreover, we conduct experiments to edit garments from several different directions, as illustrated in the second row of Fig. 9. Our method consistently demonstrates high-quality results in both inward and outward edits of clothing. It's also worth noting that, thanks to our correspondence sample design, we can achieve desirable results even when the garment features complex textures (such as the cross straps on the green sweater in the middle image). The right image in the last row demonstrates that when editing multiple layers of clothing, our method produces both accurate and aligned results according to user intent, showcasing strong generalization across different domains. More visual results are present in Appendix A.3.

### 4.5 ABLATION STUDY

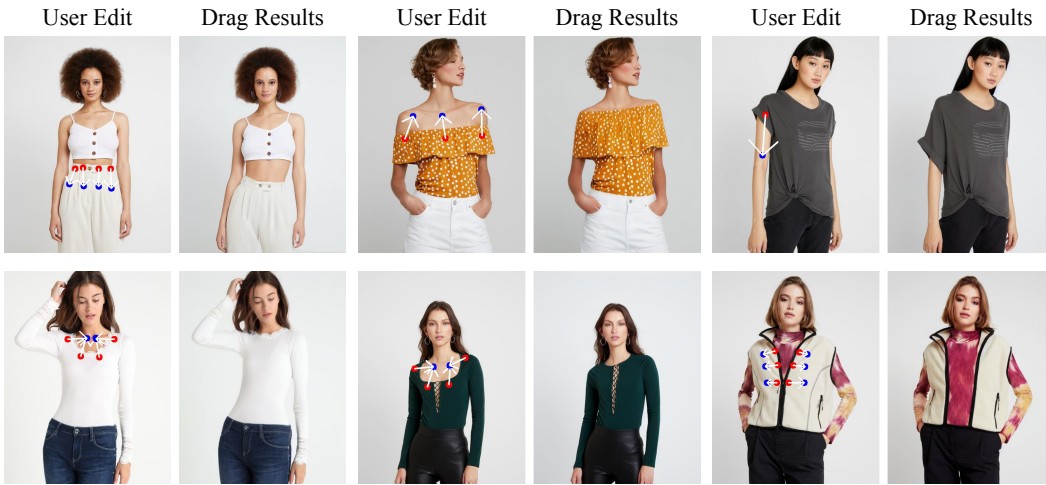

Figure 9: Visual results of the cloth editing based on the VITON-HD Choi et al. (2021) dataset. Our method achieves superior performance in modifying different parts (*e.g.*, sleeves, collars, shoulders) across various clothing types (*e.g.*, shirt, pants, jacket).

We conduct the ablation study of our approach to verify the effectiveness of each component. As illustrated in Tab. 2, we evaluate the performance of different settings based on the DragBench dataset using MD and IF metrics.

**Analysis of semantic-driven optimization** To demonstrate the effectiveness of semantic-driven latent optimization, we compare DragDiffusion with a model that only replaces the latent updating framework. As shown in the first and second rows in Tab 2, compared to the baseline DragDiffusion, the model with a semantic-driven module achieves gains of 2.71 in the MD metric and 0.088 in the IF metrics. Combined with the visual result in Fig. 5, the proposed new optimization significantly improves performance by generating high-quality results that align with user intent, facilitated by extracting more comprehensive information from the context.

Table 2: Ablation study on the two main proposed modules on the DragBench Shi et al. (2024b) dataset. The baseline method is DragDiffusion, and we replace the corresponding module in each part to assess performance.

| Semantic-Driven | CLoss | MD↓ | IF (1-LPIPS) ↑ |
|:---:|:---:|---|---|
| ✗ | ✗ | 34.29 | 0.789 |
| ✓ | ✗ | 31.58 | 0.871 |
| ✓ | ✓ | 30.69 | 0.873 |

**Analysis of Correspondence Sample** To better analyze the improvement of the sampling process, we compare the method without the corresponding loss (CLoss) in the diffusion sample stage and the proposed AdaptiveDrag, as shown in the last two rows of Tab. 2. The CLoss improves performance by 0.89 in the MD metric for precision and by 0.002 in the IF metric for feature preservation. The results demonstrate the effectiveness of CLoss in enhancing drag accuracy and preserving features.

## 5 CONCLUSION

In this paper, we proposed a novel point-based image editing method, *AdaptiveDrag*, which introduces a semantic-driven framework that offers a more user-friendly and precise drag-based editing approach compared to existing methods. With the auto mask generation module, the user can conveniently modify the images by clicking several points. Furthermore, the proposed semantic-driven optimization yields high-quality results across arbitrary dragging distances and domains. The correspondence sampling with CLoss further enhances performance by improving precision and ensuring stable feature preservation. Finally, extensive experiments demonstrate AdaptiveDrag's capability to generate images that meet user satisfaction. However, our approach still has limitations in cases of extremely long-distance dragging, where the results may not consistently with expectations. In our experiments, an improved base model version (*Stable Diffusion XL*) can broaden the manipulation range. For reproducibility, we provide our source code location and guidance in Sec. A.4.

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

# A APPENDIX

## A.1 ADDITIONAL VISUAL COMPARISION

In this section, we present additional comparisons between our AdaptiveDrag and other state-of-the-art methods, as illustrated in Fig. 10. In the first row, our method effectively rotates the black vehicle, whereas other methods show a significant loss of detail on the front of the car. Moreover, Moreover, we attempt to lower the mountain by the down-pointing arrow in the second row of Fig. 10. DragNoise does not alter the height at all, while DragDiffusion and EasyDrag fail to effectively preserve the surrounding areas. In contrast, AdaptiveDrag generates higher quality results when dragging the peak to a lower position, successfully maintaining the elements around the mountain.

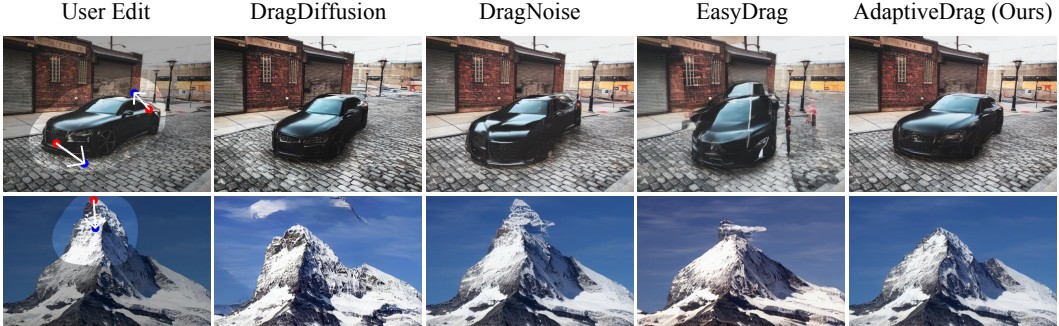

Figure 10: Additional Visual Comparison with other three state-of-art methods based on the DragBench Shi et al. (2024b) dataset. Our method also delivers more precision and high-quality results.

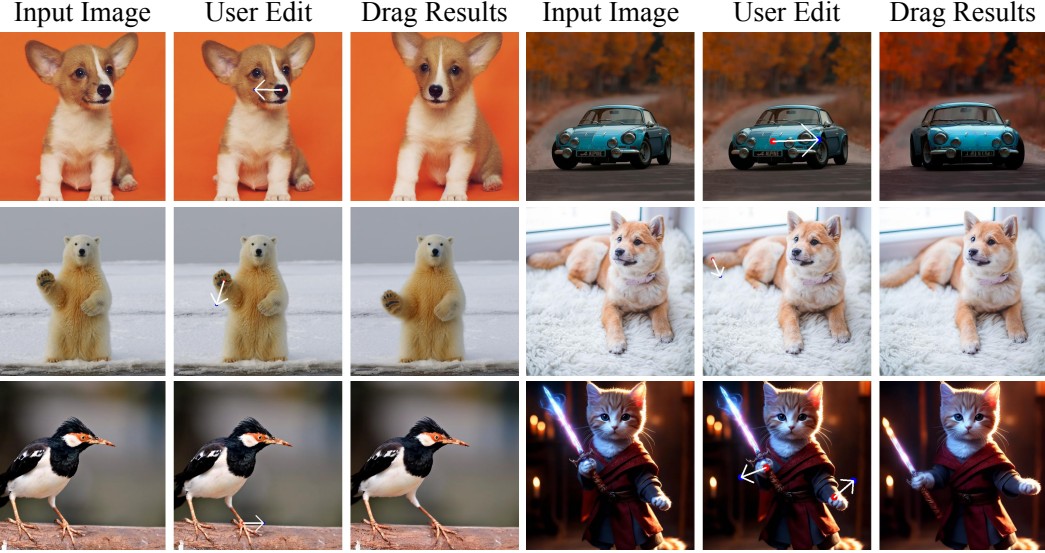

Figure 11: Visual results of the rotation and animal body part movement based on the DragBench Shi et al. (2024b) dataset.

## A.2 ADDITIONAL RESULTS OF ADAPTIVEDRAG

To verify the performance of our proposed method, we present additional visual results with various types of instructions below. The experiments in this section are conducted on the DragBench Shi et al. (2024b) dataset.

**Rotation and Movement:** As shown in Fig 11, the first row demonstrates the rotation operation, where the dog's face turns from right to left and the car changes its direction. The other two rows

| Input Image | User Edit | Drag Results | Input Image | User Edit | Drag Results |

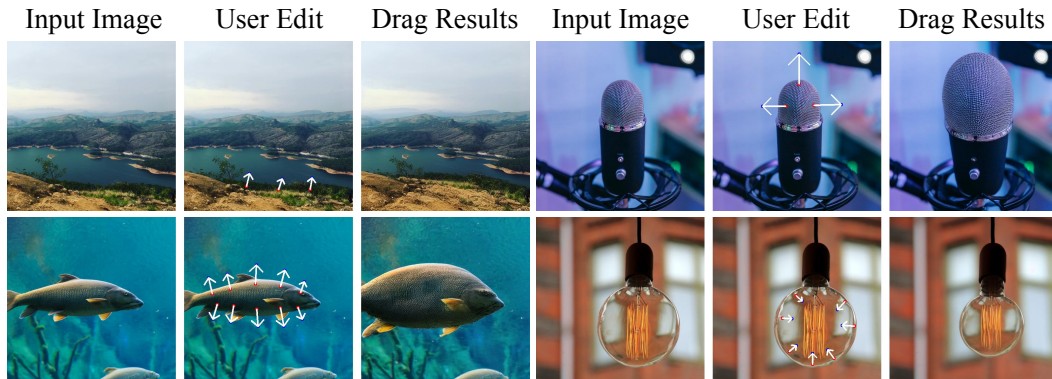

Figure 12: Visual results of the multiple points editing based on the DragBench Shi et al. (2024b) dataset.

| Input Image | User Edit | Drag Results | Input Image | User Edit | Drag Results |

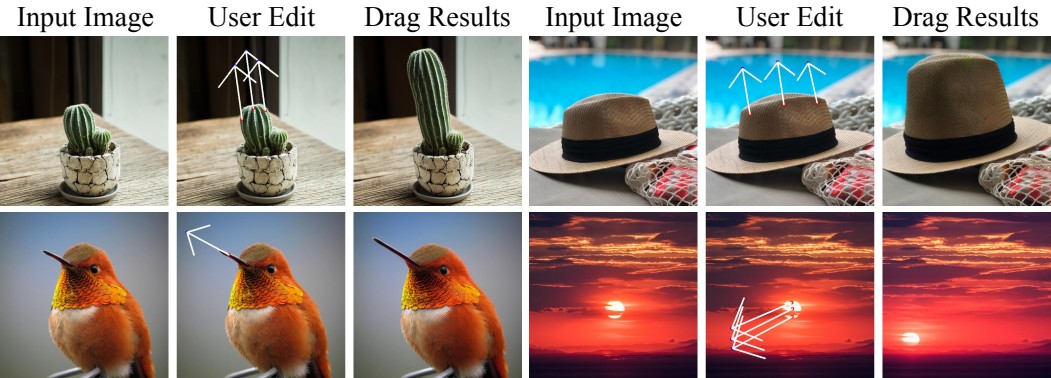

Figure 13: Visual results of long-scale editing based on the DragBench Shi et al. (2024b) dataset.

illustrate different movement operations, such as repositioning the hands, feet, or tails of animals. AdaptiveDrag demonstrates superior performance in feature retention while keeping non-dragged areas unchanged.

**Multiple Points Editing:** To enhance the drag effect, we conduct experiments on editing multiple points simultaneously, as shown in Fig. 12. In the first row, we use three points in the same direction to extend the edge of the riverside and in various directions to enlarge the microphone. In the last row, we can edit the fish at up to 10 points while maintaining dragging consistency from different locations and effectively preserving its features.

**Long-Scale Editing:** In Fig. 13, we illustrate the long-scale image editing across various scenes. Our method not only extends objects across nearly the entire image but also transforms slim items into extremely long forms. Notably, the last image demonstrates our ability to move the sun from the center to the bottom-left corner of the image.

### A.3    ADDITIONAL RESULTS OF DRAGGING INSTRUCTION ON CLOTHING

In this section, we present additional point-based image editing results for clothing using the VITON-HD Choi et al. (2021) dataset, as shown in Fig. 14. Our method also produces high-quality drag results for complex knit textures on sweaters, as shown in the left image of the first row. We can also modify different parts of a single piece of clothing using various instructions. The two groups of images in the last row demonstrate the strong generalization and adaptability of AdaptiveDrag.

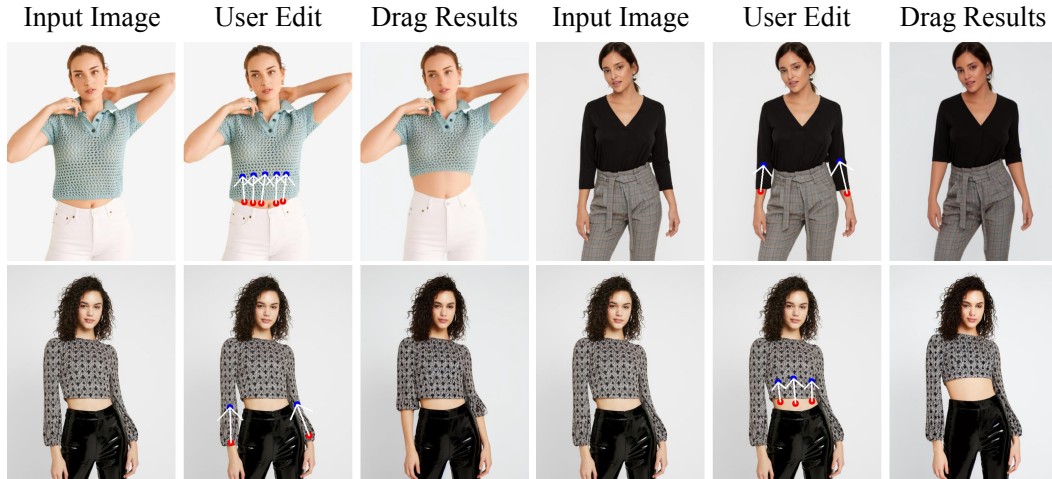

| Input Image | User Edit | Drag Results | Input Image | User Edit | Drag Results |

Figure 14: Visual results of additional clothing edits based on the VITON-HD Choi et al. (2021) dataset.

Table 3: Time consumption of DragDiffusion and AdaptiveDrag using images from the Drag-Bench Shi et al. (2024b) dataset. The experiment is conducted on a single Nvidia V100 GPU, with input images size are $512 \times 512$.

| Method | Mask | | LoRA | Optimization | Sample |
|---|---|---|---|---|---|
| AdaptiveDrag | SAM 3.0s | SLIC 0.2s | 40.1s | 10.3s | 20.4s |
| DragDiffusion Shi et al. (2024b) | 24.9s | | 39.9s | 31.2s | 5.1s |

## A.4 REPRODUCIBILITY STATEMENT

We introduce our method in this paper with four main stages: diffusion model inversion (Sec. 3.1), auto mask (Sec. 3.3)generation, semantic-driven optimization (Sec. 3.4) and correspondence sample (Sec. 3.5). Furthermore, we provide the source code in the supplementary materials to allow for a deeper understanding of the implementation of our design modules. Finally, since our method is implemented using the PyTorch framework and designed for inference on the Nvidia V100 GPU platform, it is highly reproducible, especially when combined with the detailed explanations provided in the article.

## A.5 TIME CONSUMPTION

In this section, we present the time consumption in Tab. 3. Compared with the manual mask, our auto mask network only requires 3.2s for generating a mask region. The time cost of manual methods is about 16.6s, which is obtained from the average of ten images edited by each of the five users. These results verify the efficiency of our auto mask design.

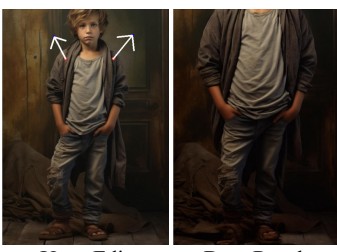

User Edit   Drag Result

## A.6 LIMITATIONS

Fig. 15 illustrates the failure case of our method. Adaptive-Drag has limitations in editing the image content in ways that do not align with real-world scenarios (*e.g.*, moving only the boy's shoulder to higher positions). This is primarily due to the pre-trained diffusion model, which incorporates basic rules that are consistent with real-world scenes.

Figure 15: A failure case of our approach. We attempt to drag the boy's shoulder to a higher position, while the entire body becomes unexpectedly expanded.

| Expand Edit | Drag Result | Moving Edit | Drag Result | Resize Edit | Drag Result |

Figure 16: Various operations in the same scene (*expand, moving, resize*) based on the Drag-Bench Shi et al. (2024b) dataset. Our method has stability with a different point-based editing.

---

**Algorithm 1:** Pipeline of AdaptiveDrag

---

**Input:** Input image $z_0$, handle point $\{p_i^0\}_{i=1}^l$, target point $\{q_i\}_{i=1}^l$, drag iterations $N$, latent time steps $T$, number of segmentation patches for SLIC $n_p$

**Output:** Output image $z_0'$

1   Finetune $U_l$ on $z_0$ with LoRA
2   Generate mask $M$ with SAM and SLIC (Sec. 3.3)
3   Superpixel segmentation patches $\{A_{(x_i,y_i)}\}_{i=1}^l$
4   $z_T \leftarrow$ DDIM inversion to $z_0$ (Eq. 3)
5   $z_T^0 \leftarrow z_T, p_i^0 \leftarrow p_i$
6   **for** *k in 0:K-1* **do**
7     $z_{T,0}^k \leftarrow z_T^k$
8     Update $z_i^k$ using motion supervision with patch $A_{(x_i,y_i)}$ as Eq. 5
9     Update $p_i^{k+1}$ using points tracking with patch $A_{(x_i,y_i)}$ as Eq. 6
10     Calculate the updating distance $d_i^u \leftarrow p_i^{k+1} - p_i^k$
11     Ideal Distance $d_i^k \leftarrow \frac{D_i}{N}$ (D is the distance from $p_i^0$ to $q_i$)
12     **if** $d_i^u < d_i^k$ **then**
13       $z_i^k \leftarrow z_i^{k-1}$
14       $p_i^{k+1} \leftarrow p_i^k$
15     **end**
16     **for** *t in T:1* **do**
17       $z_{t-1}^K \leftarrow$ denoising from $z_t^N$ (Eq. 2)
18     **end**
19     $z_0' \leftarrow z_0^K$
20   **end**

---

### A.7   STABILITY ANALYSIS

In this section, we adopt various operations in the same scene. As shown in Fig. 16, the AdaptiveDrag method achieves stable and high-quality results with expand, move, and resize operations, demonstrating the robustness and stability of our approach.

### A.8   MORE DETAILS OF ADAPTIVEDRAG

To facilitate a better understanding, we provide pseudocode for Section 3.4.2 as follows. The whole pipeline if AdaptiveDrag is present in Alg. 20. Following DragDiffusion Shi et al. (2024b), the process of the motion supervision and point tracking are provided in Eq. 5 and Eq. 6.

The motion supervision process of the latent $z_t^k$ can be formulated as:

$$\mathcal{L}_{ms}(z_t^k) = \sum_{i=1}^l \sum_{q \in A(x_i,y_i)} \left\| F_{q+d_i}(z_t^k) - \text{sg}(F_q(z_t^k)) \right\|_1 + \lambda \left\| (z_t^{k-1} - \text{sg}(z_t^{k-1})) \odot (1 - M) \right\|_1,$$

$$(5)$$

| Point Edit | Region Edit | FastDrag | InstantDrag | RegionDrag | AdaptiveDrag |
|---|---|---|---|---|---|

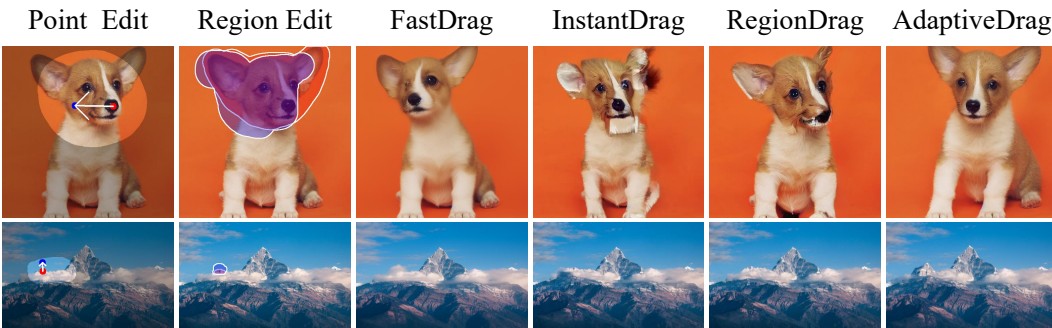

Figure 17: Additional compared methods (*FastDrag, InstantDrag, Region Drag*) based on the Drag-Bench Shi et al. (2024b) dataset. Our method also delivers more precision and high-quality results.

| User Edit | w/o CLoss | w/ CLoss | User Edit | w/o CLoss | w/ CLoss |
|---|---|---|---|---|---|

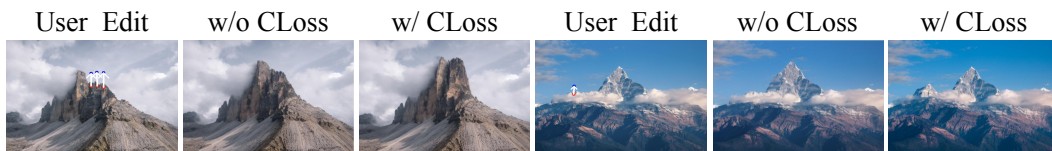

Figure 18: The visual comparison of the correspondence sample design. Specifically, 'w/o CLoss' denotes the method without CLoss in the sample stage, while 'w/ CLoss' represents our approach. AdaptiveDrag with CLoss optimization effectively edits the mountains into the desired target locations, whereas the method without CLoss fails to achieve this.

where $z_t^k$ is the $t$-th step latent after $k$-th step optimization, $sg(\cdot)$ is the stop gradient operator and $M$ is the mask region from auto mask generation network. The $F(z)$ is the output feature of the Diffusion UNet. We denote the superpixel patch centered around $p_i^k$ as $A(x_i, y_i)$,

The update process of handle points can be formulated as:

$$p_i^{k+1} = \underset{q \in A(x_i, y_i)}{\arg\min} \left\| F_q(z_t^{k+1}) - F_{p_i^0}(z_t) \right\|_1. \tag{6}$$

## A.9 ADDITIONAL METHODS COMPARISON

In this section, we present additional comparisons between our AdaptiveDrag and other state-of-the-art methods, as illustrated in Fig. 17.

## A.10 MORE ANALYSIS OF CORRESPONDENCE SAMPLE

In this section, we present the visual comparison of the "Correspondence Sample" design. As shown in Fig. 18, compared with the method without "Correspondence Sample", our AdaptiveDrag generates better quality results, which contain content more aligned with the target point locations.

