# OpenReview forum: "Adaptive Drag: Semantic-Driven Dragging on Diffusion-Based Image Editing"
_ICLR.cc/2025/Conference — Submitted to ICLR 2025_

### Official Review · Reviewer_jBPS · 2024-10-18

**Soundness:** 3
**Presentation:** 2
**Contribution:** 2
**Rating:** 5
**Confidence:** 4

**Summary:**

AdaptiveDrag introduces several innovative approaches to diffusion-based image editing, focusing on a semantic-driven optimization and auto mask generation to enhance the precision and user-friendliness of image editing tasks. The method offers advancements over existing techniques by eliminating the need for manually drawn masks and integrating semantic information into the editing process, which aligns the edits more closely with user intentions and the contextual meaning of images.

**Strengths:**

1. **Innovative Generative Mask**: The proposed method introduces an auto mask generation module that leverages semantic-driven segmentation to automatically generate precise mask areas. This approach, which facilitates image editing without user-drawn masks, represents a significant advance, potentially reducing the complexity and increasing the accessibility of point-based image editing.

2. **Semantic-Driven Optimization**: The semantic-driven optimization technique is a substantial improvement over existing methods. By incorporating semantic information into the latent optimization process, AdaptiveDrag promises to yield results that are more in line with user expectations and the semantic context of the image, improving both the precision and the aesthetic quality of the edits.

3. **Correspondence Loss**: The introduction of a correspondence loss during the sampling process is a thoughtful addition that helps in maintaining consistency between the regions of handle and target points, potentially leading to improvements in the stability of the generated images.

**Weaknesses:**

1. **Real-World Applicability and Efficiency**: While the generative mask is innovative, concerns about its practical utility are valid, especially regarding inference times. Considering that current track-and-drag methods take around 1 minute for inference, adding complexity with an additional model for mask generation might not be feasible for real-time applications. **Suggestion**: It would be beneficial to include a performance evaluation section that discusses the computational cost and inference times in comparison to manual methods and other automated approaches.

2. **Demonstration of Improvement Over Previous Methods**:
   - **Drag Missing and Feature Maintenance Failure**: The paper could strengthen its argument by providing a clearer demonstration of how it addresses these issues compared to previous methods. Using metrics like PCK0.1 to quantitatively measure the accuracy of correspondence during dragging could provide a more solid basis for claims of improvement. **Suggestion**: Enhance Figure 1 to include comparative visualizations or add a subsection that quantitatively evaluates these aspects using robust metrics.
   - **Artifacts in Edited Images**: The presence of artifacts, such as the peak of the mountain in Figure 1 and the duck's mouth in Figure 7, should be acknowledged and addressed. **Suggestion**: Discuss potential limitations in the current implementation that might lead to these artifacts and propose future work to mitigate these issues.

3. **Time Comparison**: Including a time comparison between AdaptiveDrag and other methods (e.g., DragDiffusion, EasyDrag) would provide a more comprehensive understanding of the method's efficiency. **Suggestion**: Report on both the computational time and user interaction time to give a complete picture of the method's performance in practical scenarios.

4. **Clarification of Novelty in Correspondence Loss**: If similar approaches to correspondence loss are used in methods like DragDiffusion, it is crucial to clearly differentiate AdaptiveDrag's approach and articulate the specific benefits. **Suggestion**: Provide a detailed comparison and explicitly state how AdaptiveDrag's implementation of correspondence loss improves over similar strategies in terms of accuracy, efficiency, or stability.

5. **Discussion of Related Methods**: The paper could benefit from a broader comparison with more recent and relevant techniques such as StableDrag, RegionDrag, LightningDrag, and InstantDrag. **Suggestion**: Discuss these methods in the related work section to highlight AdaptiveDrag's unique contributions and position it within the current research landscape.

6. **Addressing Similar Strategies in Other Models**: It is mentioned that SDE-Drag also drags a fixed number of pixels, similar to a point in AdaptiveDrag (Section 3.4.1). **Suggestion**: Discuss how AdaptiveDrag enhances or differs from this approach in terms of flexibility, accuracy, and user control.

**Questions:**

Could you clarify if masks are used in the DragDiffusion method to limit edits to specific image areas, and explain why there are notable changes in non-masked regions (e.g. Figure 7 Row 3)?

---

### Official Review · Reviewer_nL7N · 2024-10-27

**Soundness:** 2
**Presentation:** 3
**Contribution:** 2
**Rating:** 3
**Confidence:** 4

**Summary:**

This work proposes a mask-free point-based image editing method, which first employs SAM2 ans SLIC to generate super-pixel patches and then utlizes these local patches for semantic-driven dragging editing. Intuitively, such semantic-driven segmentation indeed provides assistance for point-based editing and helps better results in some cases. However, the patch-level constratints based on 2D segmentation will damage the 3D dragging instrcution (e.g., turns the head) as it automatically generate mask area keeping stationary. A more general condition is that point-based editing does not only affect local content, the global content of the image should also be adjusted.

**Strengths:**

1) employ local super-pixels for semantic-driven editing indeed provides assistance for point-based editing.
2) automatical  mask generation is an effective approach for avoiding unreasonable results.

**Weaknesses:**

1) Such patch-level constratints based on 2D segmentation will damage the 3D dragging instrcution (e.g., head turning)
2) The analysis regrading time consumption is missing.
3) The proposed ``corresponding sample" forces the region around  target point closely resemble the area surrounding handle point, However, such constraint may fail to provide effective guidance when handle point is far away from target point as the updated point is also forces to close to handle point.
4) The compared methods are limited, the recent approaches such as InstantDrag and FastDrag are recommended to be included.

**Questions:**

1) The implementation details of mask generation, simply move from the handle point to the target point？
2) The proposed method claims that it outperforms in long-distance edting scenarios, which design helps achieve it?

---

### Official Review · Reviewer_Gtb8 · 2024-10-28

**Soundness:** 4
**Presentation:** 3
**Contribution:** 3
**Rating:** 6
**Confidence:** 4

**Summary:**

The paper introduces a new pipeline for manipulating images using points as handles. The points can be dragged into a new position and a new image is produced in which handle points are moved to desired locations. The authors propose an optimization based approach using diffusion models to achieve this goal. Additionally, they propose a method for deriving manipulation masks automatically based on clustering segmentation features.

**Strengths:**

The proposed pipeline is sensible and carefully explained by the authors. The authors have identified many cases in which their method advances the state of the art and provide intuitions about design aspects of their work that contribute to such improvements. These are further explained through ablation studies.

The Semantic Region approach in sec 3.4.2 is an interesting approach to solving the problem of mixing regions and has clear demonstrable benefits.

**Weaknesses:**

1) It feels to me that the method is underspecified.
a) For one the definition of a mask by the user should provide information about their intent but here it is automatically generated by the model.
b) It is not clear how the user can distinguish between moving an object, extending an object and other operations. In manuscript most examples show extending an object by stretching. Is there a way to constraint the model for specific editing operations in the same scene? Making use of text could disambiguate these tasks.

2) In terms of text comprehension, reading 3.3 and Fig.3 it is not clear to me how the superpixels are chosen to construct a mask. Clearly height is defined by source to target distance and all superpixels along the way are selected, but what about the other direction? Also many superpixels below source point are shown as selected in Fig3d.

Not exactly weaknesses:
2) Similarly to SOTA competing methods, the proposed pipeline involves many complex operations run in an optimization setting (many forward passes are required) which hints to very long runtimes for inference use. A runtime analysis and comparison with literature is essential information here.

typos:
- p. "We prose"
- fig.2 both VAE read "VAE Encoder"

**Questions:**

Weaknesses:
1b) Is there a way to constraint the model for specific editing operations in the same scene?
2) It is not clear how chosen, please elaborate on that.
3) what are the memory requirements and runtime during inference?

---

### Official Review · Reviewer_9jL2 · 2024-10-31

**Soundness:** 2
**Presentation:** 3
**Contribution:** 3
**Rating:** 6
**Confidence:** 4

**Summary:**

This paper proposes AdaptiveDrag, a mask-free, semantic-driven image editing method. It employs super-pixel segmentation for accurate auto mask generation and uses adaptive, semantic-driven optimization. Additionally, a correspondence sample ensures feature consistency.

**Strengths:**

1.	This method eliminates the need for user-input masks by generating precise, semantic-based masks automatically.
2.	This paper produces visually high-quality and accurate editing results.

**Weaknesses:**

1.	Some sections, particularly Section 3.4.2, involve complex processes that could benefit from algorithmic pseudocode or flowcharts to enhance clarity. This would make it easier for readers to understand the step-by-step workings of the method.
2.	This paper lacks visual evidence for Correspondence Sample. While the paper introduces the Correspondence Sample for feature consistency, it lacks visual results to demonstrate its effectiveness.

**Questions:**

See the weaknesses. Another point worth considering is whether using regional optimization in AdaptiveDrag could reduce the number of required edit points. By focusing on broader regions, it might achieve desired edits more efficiently, simplifying the process and enhancing usability.

---

### Official Review · Reviewer_9ED8 · 2024-11-02

**Soundness:** 2
**Presentation:** 3
**Contribution:** 2
**Rating:** 5
**Confidence:** 3

**Summary:**

This paper presents a  mask-free point-based image editing approach called AdaptiveDrag. This method enhances the quality and precision of interactive image editing by leveraging semantic information in diffusion models. It consists Auto Mask Generation,
Semantic-Driven Optimization, and  Correspondence Sampling. Experiments are conducted on several drag instructions.

**Strengths:**

1. The paper presents a well-designed semantic-driven optimization
2. The paper compared with the most existing methods to demonstrate the performance in both qualitative and quantitative metrics.
3. The paper is well-organized.

**Weaknesses:**

1.  While the auto mask generation can provide the fine-grained patches, its stability is not evaluated.
2. The computational cost should be evaluated since the setup requires super-pixel segmentation using SAM and SLIC, which may add computational overhead. Moreover, this method is also based the LoRA tuning.
3. While the paper acknowledges issues with long-distance dragging, it does not show and discuss failure cases  for better understanding the model.

**Questions:**

See the strengths and weaknesses

---

### Meta-Review · Area_Chair_zpCP · 2024-12-19

**Metareview:**

This submission introduces a point-based method that leverages semantic information to improve the quality and precision of interactive image editing. The approach attempts to align edits with contextual meaning and to ensure feature consistency. This enables users to interactively drag points to desired positions for local and global image content adaptation. Experimental work assesses quantitative and qualitative edit quality in comparison with recent approaches.


After reviewing the paper, rebuttal and resulting discussions the consensus was that although the review process has helped to strengthen the work, valid open reviewer concerns remain. A majority of reviewers are unconvinced post rebuttal and lean towards rejection. AC believes this work shows visually promising results yet can likely benefit from a further round of review and therefore recommends rejection. Authors are encouraged to use the collective feedback to strengthen the work towards submission at an alternative venue.

**Additional Comments On Reviewer Discussion:**

The paper receives five reviews resulting in: two borderline accepts, two borderline rejects and one clear reject.

A subset of reviews comment on positive aspects, highlighting a well-designed method and exposition that can link technical components to visual improvements. Some reviewers note that introduced semantic-driven optimisation and regions, correspondence loss components can lead to useful benefits; namely the removal of user-drawn mask requirements and improved aesthetics. Further positive points speak to appropriate comparisons, a well-organised paper and high-quality visual results.

Reviews also raised a somewhat wide-ranging (and in cases contrary) set of problematic issues, highlighting: a lack of evaluations (stability, computational cost and runtime analysis) and insufficient exposition of complex parts. More fundamental concerns considered the potential problems of an underspecified method, the intrinsic limitations of 2D methods with respect to edits that entail 3D semantics and logical concerns with the proposed correspondence constraints. Additional queries and issued arose due to initially unclear visual examples, unsupported claims, missing failure cases and missing comparisons.

The rebuttal period observed that author responses could resolve a subset of concerns through somewhat limited discussion and direct updates to the manuscript. A subset of reviewers note that fundamental concerns explicitly remain open relating to the under-constrained nature of the model and a persisting lack of clarity in the exposition (namely super-pixel selection algorithm). Follow-up concerns and queries probed during the rebuttal included limitations for 3D content edits, inconsistencies in time comparisons, missing quantitative results, feature preservation strategy, and the unreasonableness of region-level alignment constraints during early dragging stages. AC considers these concerns valid yet they unfortunately appear to remain largely unaddressed.

---

### Decision · Program_Chairs · 2025-01-22

Reject